# Edible Microalgae and Their Bioactive Compounds in the Prevention and Treatment of Metabolic Alterations

**DOI:** 10.3390/nu13020563

**Published:** 2021-02-09

**Authors:** Sara Ramos-Romero, Joan Ramon Torrella, Teresa Pagès, Ginés Viscor, Josep Lluís Torres

**Affiliations:** 1Physiology Section, Department of Cell Biology, Physiology and Immunology, Faculty of Biology, University of Barcelona, 08007 Barcelona, Spain; jtorrella@ub.edu (J.R.T.); tpages@ub.edu (T.P.); gviscor@ub.edu (G.V.); 2Department of Biological Chemistry, Institute of Advanced Chemistry of Catalonia (IQAC-CSIC), 08034 Barcelona, Spain; joseplluis.torres@iqac.csic.es

**Keywords:** *Spirulina* (*Arthrospira*), *Chlorella*, *Isochrysis*, *Tetraselmis*, *Nannochloropsis*, algae, omega 3, obesity, diabetes

## Abstract

Marine and freshwater algae and their products are in growing demand worldwide because of their nutritional and functional properties. Microalgae (unicellular algae) will constitute one of the major foods of the future for nutritional and environmental reasons. They are sources of high-quality protein and bioactive molecules with potential application in the modern epidemics of obesity and diabetes. They may also contribute decisively to sustainability through carbon dioxide fixation and minimization of agricultural land use. This paper reviews current knowledge of the effects of consuming edible microalgae on the metabolic alterations known as metabolic syndrome (MS). These microalgae include *Chlorella*, *Spirulina* (*Arthrospira*) and *Tetraselmis* as well as *Isochrysis* and *Nannochloropsis* as candidates for human consumption. *Chlorella* biomass has shown antioxidant, antidiabetic, immunomodulatory, antihypertensive, and antihyperlipidemic effects in humans and other mammals. The components of microalgae reviewed suggest that they may be effective against MS at two levels: in the early stages, to work against the development of insulin resistance (IR), and later, when pancreatic -cell function is already compromised. The active components at both stages are antioxidant scavengers and anti-inflammatory lipid mediators such as carotenoids and -3 PUFAs (eicosapentaenoic acid/docosahexaenoic acid; EPA/DHA), prebiotic polysaccharides, phenolics, antihypertensive peptides, several pigments such as phycobilins and phycocyanin, and some vitamins, such as folate. As a source of high-quality protein, including an array of bioactive molecules with potential activity against the modern epidemics of obesity and diabetes, microalgae are proposed as excellent foods for the future. Moreover, their incorporation into the human diet would decisively contribute to a more sustainable world because of their roles in carbon dioxide fixation and reducing the use of land for agricultural purposes.

## 1. Introduction

The main risk factors for developing cardiovascular disease (CVD), which is the leading cause of death worldwide, are type 2 diabetes mellitus (T2DM) and obesity, together with hypertension and hypercholesterolemia. Diabetes is a pathology characterized by hyperglycemia resulting from a total or partial lack of insulin. While type 1 diabetes (T1DM), also called insulin-dependent diabetes mellitus, is due to autoimmune destruction of insulin-producing pancreatic β-cells, T2DM, also called insulin-independent diabetes mellitus, is caused by insulin resistance (IR) in tissues (mainly adipose tissue, liver and muscle) followed by a failure of β-cells to compensate for this [1]. IR and T2DM are usually associated with excessive ingestion of saturated fat and refined sugar [2,3]. Obesity, IR and T2DM usually occur together with a state of low-grade systemic inflammation which may be both the cause and a consequence of these metabolic alterations [4]. Hypertension is another crucial CVD risk factor that may be triggered by a poor diet (excess salt, fat, or fructose) [5]. Hypercholesterolemia completes the list of CVD risk factors, as it promotes atherosclerotic plaque formation [6]. The cluster of factors leading to T2DM and CVD is called metabolic syndrome (MS), a diagnostic concept used in clinical practice [7]. Obesity, T2DM, hypercholesterolemia, and, to some extent, hypertension are modifiable risk factors for CVD as they may be controlled/delayed by adopting a healthy lifestyle that includes a balanced diet and physical activity.

Some functional foods and bioactive components isolated from them can potentially be used as tools to prevent CVD risk factors [8]. The consumption of several species of microalgae produces health benefits in humans and other animals [9]. In this paper, we review the current evidence that supports the benefits of consuming edible microalgae in relation to diet-induced metabolic alterations. These edible microalgae include *Arthrospira* (*Spirulina*), *Chlorella*, and *Tetraselmis*. We have also included *Isochrysis galbana* and *Nannochloropsis* because of their potential to act as metabolic health-promoting functional foods in view of both their capacity for storage of functional oils and their extensive history of use in aquaculture, making them candidates for human consumption.

## 2. Microalgae

Algae are mostly photosynthetic organisms and include eukaryotic and prokaryotic species that occur in fresh and salt water. Algae belong broadly to eleven major phyla: Charophyta, Chlorarachniophyta, Chlorophyta, Cryptophyta, Cyanophyta, Dinophyta, Euglenophyta, Glaucophyta, Haptophyta, Heterokontophyta, and Rhodophyta. Microalgae are unicellular algae that are between 1 and 50 µm in diameter and comprise a highly diverse group of 200,000–800,000 species [10].

Microalgae are dense in protein; some species even contain similar amounts of protein to those found in milk, eggs, and meat [9,11]. Microalgae also contain several bioactive components with therapeutic potential, such as dietary fiber, polyphenols, carotenoids, phycobiliproteins, polysaccharides, vitamins, sterols, and, particularly, polyunsaturated fatty acids (PUFAs) such as the ω-3 PUFAs eicosapentaenoic acid (EPA, 20:5 n-3) and docosahexaenoic acid (DHA, 22:6 n-3) (Table 1) [11,12,13]. The proportion of bioactive compounds varies between species and also depends on growing conditions (mainly temperature, illumination, pH, CO_2_ supply, salt, and nutrients) [14]. *Arthrospira* (also known as *Spirulina*), *Chlorella*, and *Tetraselmis* are currently used for human nutrition. The European Commission (EC) only includes *Arthrospira* and *Chlorella* in their novel food catalogue. The currently preferred name for the genus *Spirulina* is *Arthrospira*. In this paper we use the term *Arthrospira* when referring to the genus and spirulina when referring generically to biomass preparations of the microalga. We will also use the terms spirulina and chlorella when referring to studies in which the particular species used is not specified. (https://ec.europa.eu/food/safety/novel_food/catalogue/search/public/index.cfm, accessed on 15 August 2020).

*Arthrospira* is actually a cyanobacteria considered to be a blue-green microalga that has historically been consumed by North Africans and Mexicans because of its nutritional value, containing 60–70% protein by dry weight (Table 1) and bioactive compounds [15]. *Arthrospira* species are abundant in tropical and subtropical areas with carbonate and bicarbonate-rich alkaline water bodies [10]. They contain high concentrations of antioxidants (β-carotene and phycocyanin), minerals (K, Na, Ca, Mg, Fe, Zn), vitamins (tocopherols), eight essential amino acids, PUFAs (especially γ-linolenic acid (ALA, 18:3 n-6)), and phenolic compounds [15]. Nowadays, spirulina is used as a nutritional dietary supplement, mainly due to its anti-inflammatory activity, and its intake is recommended for individuals with pathologies and conditions such as arterial hypertension, IR and diabetes among others.

Algae belonging to the *Chlorella* genus live in both fresh and salt water and use metabolic pathways similar to higher plants. *Chlorella* belongs to the phylum Chlorophyta [10]. These microalgae have a high protein content (Table 1) that includes the essential amino acids isoleucine, leucine, lysine, methionine, phenylalanine, threonine, tryptophan, valine, and histidine [16], along with minerals, vitamins, and carotenoids. They also contain dietary fiber and chlorophyll [17]. *Chlorella* can also synthesize large amounts (as much as 50% dry weight) of triacylglycerols (TAG) under stressful conditions (e.g., exacerbated light or nitrogen deficiency) [18]. The overall composition of different *Chlorella*-derived products is 59–70 g protein, 5–20 g fat, 20% carbohydrates, and 5–18 g fiber [16].

## 3. Spirulina (*Arthrospira*) and Metabolic Alterations

In vivo studies indicate that *Arthrospira maxima* and *platensis*, as well as other microalgae, exert their anti-obesity effects via the reduction of both adipogenesis in white adipose tissue (WAT) and lipogenesis in WAT and brown adipose tissue (BAT). They increase lipolysis in WAT, lipid oxidation in WAT and skeletal muscle, and also thermogenesis and mitochondriogenesis in WAT, BAT, and skeletal muscle [19]. An ethanolic extract of *A. maxima* (150 or 450 mg/kg/day) reduced body weight, both subcutaneous and visceral adipose tissue, blood fasting glucose and lipid concentrations in mice fed a high-fat (HF) diet [20]. These changes were associated with lower protein expression of factors related to adipogenesis and higher expression of proteins related to adenosine 5′-monophosphate-activated protein kinase-α (AMPKα)-induced adipose browning [20]. In rats, the administration of dried *A. maxima* (62.5, 125, or 250 mg/kg) also reduced weight gain and the elevated WAT index induced by an HF diet, and it attenuated the changes related to metabolic alterations, including serum adiponectin, leptin, tumor necrosis factor α (TNF-α), glucose, insulin, and the lipid profile. These effects of *A. maxima* appear to be associated with activation of the AMPK pathway and sirtuin 1 (SIRT1) in mesenteric adipose tissue and skeletal muscle, leading to the suppression of lipid synthesis [21].

Another species, *A. platensis*, modulates dysbiosis, intestinal inflammation, and gut permeability in rats fed an HF diet. When administered as 3% of feed, it counteracted the dysbiotic changes triggered by the HF diet, namely the increased populations of Proteobacteria and Firmicutes. *A. platensis* also decreased inflammatory cytokines and the expression of myeloid differentiation factor 88 (MyD88), toll-like receptor 4 (TLR4), and NF-κB p65, as well as that of tight junction proteins in the intestinal mucosa (ZO-1, Occludin, and Claudin-1) [22]. A recent meta-analysis of 12 clinical trials analyzed the effect of spirulina supplementation on anthropometric indexes [23]. Spirulina was found to reduce body weight and waist circumference as well as body mass index when supplementation lasted for more than 12 weeks. The authors therefore suggest that spirulina may be used as an adjuvant treatment for obesity [23].

Spirulina biomass as well as the different extracts obtained from it have shown potential as antidiabetic agents. While studies that focus on the prevention of diabetes are scarce, a recent review summarized studies in which spirulina was tested in humans presenting different MS factors [24]. In one study, ingestion of 2–6 g of spirulina per day resulted in an improvement in insulin sensitivity and a reduction in glycated hemoglobin (HbA1c), although other studies did not show any detectable effect [24]. Studies in animal models have also shown an effect of spirulina on metabolic risk factors. *A. platensis* was found to counteract hyperglycemia and hyperlipidemia induced by alloxan in mouse [25] and rat [26,27] models of T1DM [28]. Moreover, *A. platensis* (5% in the diet) counteracted renal injury and oxidative stress in alloxan-induced diabetic rats [29]. *A. platensis* also showed antidiabetic effects in streptozotocin (STZ)-injected rats [30,31,32]; animals injected with STZ are also models of T1DM [28]. *A. platensis* (500 mg/kg body weight, 2 months) significantly decreased serum glucose, HbA1c, and malondialdehyde (MDA) levels and significantly increased the serum insulin concentration and the activity of antioxidant enzymes, as well as normalizing their mRNA gene expression and inducing upregulation of the gluconeogenic enzyme pyruvate carboxylase (PC), the pro-apoptotic factor Bax and caspase-3 (CASP-3), and TNF-α gene expression [31]. The authors suggested that the antioxidant, anti-inflammatory, and anti-apoptotic properties of spirulina might be due to its polyphenolic components. In an HF diet/low-dose STZ (HFD/STZ) rat model of diabetes, oral doses of *A. platensis* (250, 500 or 750 mg/kg body weight) for 30 days were shown to ameliorate levels of fasting blood glucose, insulin, and hepatic enzymes [32]. *A. platensis* also influenced the serum lipid profile and exhibited an anti-inflammatory effect via TNF-α and adiponectin modulation, in turn, probably mediated by the sterol regulatory element-binding transcription factor-1c (SREBP-1c) [32].

Arthrospyra contains a variety of bioactive components that may contribute to its beneficial effects on diabetes-associated alterations (hyperglycemia, hyperlipidemia, inflammation, and oxidative stress) acting through different mechanisms. The biomass of a typical industrial preparation of spirulina contains 71.7 g protein, 8.5 g fat, 3.0 g fiber, 16.2 g phycocyanin, and 477.0 mg carotenoids per 100 g dry weight [33]. It has been suggested that the dietary fiber and bioactive peptides are primarily responsible for the protection against IR it provides [34].

### 3.1. Dietary Fiber from Spirulina

Dietary fiber is believed to prevent IR through maintaining balanced gut microbiota (prebiotic effect) and its direct action on epithelial and immune cells that regulate the intestinal barrier and immune function [35]. Spirulina (*A. platensis)* biomass has been shown to promote the growth of putatively beneficial microorganisms (e.g., *Lactobacillus casei*, *L. acidophilus*, *Saccharomyces thermophilus*, and *Bifidobacterium* spp.) and to reduce the populations of putatively harmful bacteria in vitro [36,37,38]. In healthy male mice, spirulina (1.5–3.0 g of spray-dried *A. platensis* powder/kg body weight daily for 24 days) was found to modify the cecal populations of microbiota at the genus level (*Clostridium* XIVa, *Desulfovibrio*, *Eubacterium*, *Barnesiella*, *Bacteroides*, and *Flavonifractor*). These changes correlated with markers of oxidative stress and with blood lipid levels [39,40]. A polysaccharide isolated from spirulina was effective at lowering blood glucose and increasing superoxide dismutase (SOD) in STZ-induced diabetic Sprague-Dawley rats (100–200 mg/kg body weight by intragastric administration for 8 weeks) [41]. Whether the effects of spirulina are partially mediated by its fiber components (e.g., oligo and polysaccharides) remains to be clarified.

### 3.2. Peptides and Enzymes from Spirulina

Hydrolysates from seaweeds and microalgae contain bioactive peptides with putative applications in the food industry [42,43,44]. Peptides from spirulina biomass may make a decisive contribution to its antidiabetic effects. Proteins in *A. platensis* mainly consist of two phycobiliproteins (PBP): C-phycocyanin (C-PC) and allophycocyanin (APC) [45]. Tripsin hydrolysates of PBP yield fragments with dipeptidyl peptidase-IV (DPP-IV) inhibitory activity [46]. DPP-IV is a serine exopeptidase that is considered a promising target for the management of T2DM, because it plays a key role in glucose metabolism via N-terminal truncation and subsequent inactivation of the incretins glucagon-like peptide 1 (GLP-1) and gastrointestinal insulinotropic peptide (GIP), which are responsible for most postprandial insulin secretion [47]. The peptide Leu-Arg-Ser-Glu-Leu-Ala-Ala-Trp-Ser-Arg obtained from *A. platensis* by ultrasound treatment and subcritical water extraction exhibited inhibition of DPP-IV (IC_50_ 167.3 μg/mL). This peptide also showed further activity that would contribute to the protective effects of *A. platensis* against hyperglycemia: inhibition of α-amylase (IC_50_ 313.6 μg/mL) and α-glucosidase (IC_50_ 134.2 μg/mL) [48]. These two activities are important because they delay the digestion of starch and, consequently, lower post-prandial glycaemia [49]. While the potency of this decapeptide against these three enzymes is modest, the combined effect of these and other activities (e.g., the prebiotic effects of dietary fiber) may result in an efficient overall effect for the whole biomass. Enzymatic hydrolysates of spirulina biomass also contain angiotensin I-converting enzyme (ACE-I) inhibitors [50]. ACE-I is a dipeptidyl carboxypeptidase that catalyzes the conversion of angiotensin I to angiotensin II, a process that increases blood pressure. ACE-I inhibitors reduce the concentration of angiotensin II and consequently lower blood pressure [51]. This antihypertensive effect of peptidic fractions (200 mg/kg body weight) in spontaneously hypertensive rats has been attributed to the active peptides in *A. platensis*: Ile-Ala-Glu (IC_50_ 34.7 µM), Phe-Ala-Leu, Ala-Glu-Leu, Ile-Ala-Pro-Gly (11.4 µM), and Val-Ala-Phe (35.8 µM) [52]. Furthermore, the decapeptide Gly-Ile-Val-Ala-Gly-Asp-Val-Thr-Pro-Ile from *A. platensis* has been found to exert direct endothelium-dependent vasodilation ex vivo via a PI3K (phosphoinositide-3-kinase)/AKT (serine/threonine kinase Akt) pathway, resulting in NO release [53].

### 3.3. Unsaturated Fatty Acids from Spirulina

The ethanolic (95% ethanol) fraction of *A. platensis* biomass contains a mixture of unsaturated fatty acids that have a hypolipidemic effect in Wistar rats fed an HF diet [54]. This effect is mediated via upregulation of the AMPK-α pathway and downregulation of the SREBP-1c and the 3-hydroxy-3-methyl glutaryl coenzyme A reductase, acetyl CoA (HMG-CoA) pathways in the liver. The extract was found to increase the populations of putatively beneficial bacteria, such as *Prevotella*, *Alloprevotella*, *Porphyromonadaceae*, *Barnesiella*, and *Paraprevotella*, while reducing the populations of *Turicibacter*, *Romboutsia*, *Phascolarctobacterium*, *Olsenella*, and *Clostridium* XVIII, which correlated positively with serum TAG, total cholesterol (TC), and low-density lipoprotein cholesterol levels, but negatively with serum high-density-lipoprotein TC levels [54]. The ethanolic (55% ethanol) fraction (SP55) extracted from *A. platensis* showed antihyperglycemic activity in male rats fed an HF diet, as assessed by the oral glucose tolerance test (OGTT) [55]. The extract contained both saturated and unsaturated fatty acids. Other active components, such as polyphenols and peptides, may also have been extracted under the conditions used. The SP55 fraction appears to increase the gut populations of *Oscillibacter*, *Parasutterella*, and *Alloprevotella* and to decrease the abundance of *Turicibacter* [55].

### 3.4. Polyphenols from Spirulina

Phlorotannins (phloroglucinol-based polyphenols) and bromophenols (brominated phenolic derivatives) are both families of polyphenols that are abundant in algae [43]. Algal polyphenols have shown small-to-medium positive effects on fasting blood glucose, TC, and low-density lipoprotein (LDL) cholesterol levels in humans [43]. Little is known about the possible role of polyphenolic components in *Arthrospyra* in its antidiabetic effect. A polyphenol-rich butanol extract was found to have quite potent α-glucosidase inhibitory activity (IC_50_ 23 μg/mL) [56]. Inhibitors of intestinal α-glucosidases are instrumental in the management of diabetes, because they lower postprandial blood glucose levels. The total phenolic and flavonoid contents were estimated to be 121 mg gallic acid equivalents/100 g and 27 mg rutin equivalents/100 g *A. platensis* biomass, respectively, but no more information was provided on the structure of the putative phenolics [56]. Another ethanolic extract of spirulina biomass obtained after hydrolysis showed α-amylase inhibitory activity [57]. Although the possible structures of the active polyphenols were not revealed, the authors showed evidence that chlorogenic acid is a major component in the extract.

### 3.5. Pigments from Spirulina

Phycobilins are secondary pigments in microalgae that capture light energy while protecting microalgae from harmful radiation [58]. Phycocyanin, a blue pigment biosynthesized by *Arthrospyra*, was found to protect insulin-producing pancreatic islets from alloxan injury in mice at doses of 100 and 200 mg/kg body weight [59]. It also reduced fasting blood glucose and glycosylated serum protein (GSP) levels, maintained the total antioxidative capacity, reduced TC levels and TAG levels in the serum and liver, increased the level of hepatic glycogen, and maintained glucokinase (GK) expression in the liver. The authors suggested that inhibition of the p53 pathway could be one of the mechanisms responsible for the protection provided by phycocyanin, as it decreased p53 expression in the pancreas at the mRNA level [59]. Phycocyanin may also exert its antidiabetic effect via the inhibition of both α-amylase and α-glucosidase, as suggested by molecular docking and in vitro testing [60]. Moreover, phycocyanin from *A. platensis* reduced plasma TC and LDL cholesterol as well as oxidative stress and NADPH oxidase expression induced by an atherogenic diet in hamsters, particularly when administered together with selenium [61]. The authors suggested that phycocyanin might prevent atherosclerosis.

β-Carotene extract obtained from *A. platensis* biomass presented antihyperglycemic activity in STZ-induced diabetic mice when given at a dose of 100 mg/kg body weight after 10 days of treatment [62].

## 4. *Chlorella* and Metabolic Alterations

Chlorella supplementation in humans and other mammals has been shown to have antioxidant, antidiabetic, immunomodulatory, antihypertensive, and antihyperlipidemic effects [16,63]. Chlorella was found to improve fat metabolism in subjects with a high-risk lifestyle, together with producing significant reductions in total blood serum TC, high-density lipoprotein (HDL) cholesterol, and LDL cholesterol [64]. Studies with animal models also revealed the role of *Chlorella* products in restoring normal lipid levels [64]. *C. vulgaris* reduced serum TC, low-density lipoprotein cholesterol, and TAG in dyslipidemic subjects [65]. Chlorella (3 g/day for 4 weeks) decreased arterial stiffness in middle-aged and older individuals, together with producing an increase in NO production by the vascular endothelium [66]. The large amount of arginine in *Chlorella* proteins (3.2 g per 100 g dry weight) [16] could explain this effect.

In patients suffering from NAFLD, *C. vulgaris* was found to lower body weight and increase the serum insulin concentration and the HOMA-IR (homeostatic model assessment for IR) score, while the levels of serum glucose and TNF-α after treatment were significantly different between groups [67]. This suggests that *C. vulgaris* supplies anti-inflammatory agents capable of reverting damage to pancreatic β-cells. Chlorella also reduced body fat, serum TC, and fasting blood glucose levels in subjects with high-risk lifestyles (borderline fasting blood glucose, glucose tolerance, TC and TAG) [64]. The authors showed that a chlorella preparation obtained by crushing and spray-drying modifies the expression of genes involved in the activation of insulin signaling pathways in peripheral blood cells [64]. Chlorella intake (8.0 g/day) was found to reduce the expression of resistin (an IR inducer) in peripheral blood cells of borderline diabetics; resistin mRNA expression significantly correlated with changes in levels of HbA1c and the inflammation markers TNF-α and IL-6 [68]. In some other clinical studies, glucose metabolism appeared not to be affected by preparations of *C. vulgaris*, such as in those involving dyslipidemic subjects (dose: 600 mg/day) [65].

The mechanisms by which chlorella might exert protection against diabetes and its risk factors in humans are largely unknown, but studies using animal models have provided some information on this. An early study on alloxan-induced T1DM in Wistar rats showed that intraperitoneally-injected *C. pyrenoidosa* swiftly counteracted hyperglycemia without affecting insulin secretion [69]. In agreement with a previous paper [70], the authors concluded that the action of the injected chlorella consisted of consuming circulating glucose. In agreement with those results, orally administered chlorella (*C. pyrenoidosa*, 100 mg/kg body weight) was not shown to affect the basal blood glucose level in STZ-induced diabetic mice [71]. As supplementation prolonged the hypoglycemic effect of injected insulin, the authors suggested that chlorella may foster insulin sensitivity [71]. In another study using the STZ-model of T1DM, the authors suggested that the hypoglycemic effect of an unspecified dose of suspended chlorella powder is due to enhanced glucose uptake in the liver and soleus muscles, as ascertained by an increased uptake of 2-deoxyglucose in both normal and STZ mice [72]. The authors also suggested that the improved insulin sensitivity might be connected with reduced levels of non-esterified fatty acids, as this particular type of lipid has been linked to impaired insulin signaling [72]. In another study, glucose-stimulated insulin secretion was not affected by the intake of chlorella (*C. vulgaris*), while it improved insulin sensitivity in type 2 diabetic Goto-Kakizaki (insulin resistant with impaired β-cell function [73]) and normal Wistar rats [74]. Chlorella powder (25, 50, and 100 mg/kg body weight) also showed the capacity to improve insulin sensitivity in male Wistar rats with fructose-induced IR [75]. There seems to be a consensus in the literature that chlorella biomass improves insulin sensitivity.

Several patent applications claim that *Chlorella*-derived products, including polysaccharides, have antidiabetic actions [76,77,78,79]. Bioactive components may contribute to the effects of *Chlorella* in preventing metabolic alterations (hyperglycemia, hyperlipidemia, inflammation and oxidative stress) through different mechanisms (Table 1).

### 4.1. Unsaturated Fatty Acids from Chlorella

The ethanolic (55% ethanol) fraction (CP55) extracted from *C. pyrenoidosa* showed antihyperglycemic activity in male rats fed an HF diet, as assessed by the OGTT. CP55 was found to be more effective than the 55% ethanol fraction extracted from *S. platensis* [55]. The former extract contained both saturated and unsaturated fatty acids. Other active components, such as polyphenols and peptides, may have been extracted under the conditions used. CP55 increased the abundance of *Ruminococcus*, *Parasutterella*, and *Erysipelotrichacea* and decreased the abundance of *Lactobacillus*, *Turicibacter*, and *Blautia* [55].

### 4.2. Polysaccharides from Chlorella

Polysaccharides may be partially responsible for the antihyperlipidemic and antihyperglycemic activity of chlorella biomass via the promotion of gut eubiosis (balanced microbiota populations). *Chlorella* spp. contain β-glucans (polymers of β-D-glucose linked through 1–3 β-glycosidic bonds, 6–9% of dry weight) [80], which may very well contribute to the overall effects of *Chlorella* products. Microbial exopolysaccharides (e.g., curdlan, dextran, gellan, glucans, hyaluronic acid, levan, and pullulan) can reduce inflammatory responses by promoting gut microbiota balance, strengthening intestinal barrier function, enhancing antioxidant activities, promoting short-chain fatty acid (SCFA) production, and reducing the concentrations of pro-inflammatory mediators [81]. β-Glucans promote the growth of probiotic *Lactobacillus* and *Bifidobacterium* as well as the SCFAs propionic and butyric acid, which has been related to protection against IR and other risk factors [82]. Phosphoric acid hydrolysates of chlorella and spirulina generate oligosaccharides with potential prebiotic activity, as they promote the growth of *Bifidobacterium animalis* and *Lactobacillus casei* in vitro [83,84]. Arabinomannans (oligomers of arabinose and mannose) are components of the cell wall of *C. vulgaris* that may show potential as prebiotics [85]. A polysaccharide from *C. pyrenoidosa* increased the populations of *Coprococcus*, *Lactobacillus*, and *Turicibacter*, whereas it reduced those of the *Ruminococcus gauvreauii* group in HF-fed Wistar rats at doses of 150 and 300 mg/kg body weight. The monomer constituents are mannose, rhamnose, glucose, fucose, xylose, and arabinose in the molar ratio 14.95:13.75:11.42:10.35:4.95:3.63. This polysaccharide also contains glucuronic acid (5.5%) and has a hypolipidemic effect [86].

### 4.3. Peptides from Chlorella

Some peptides released from the microalgae by hydrolytic treatments are free radical scavengers with antioxidant properties. These peptides include Val-Glu-Cys-Tyr-Gly-Pro-Asn-Arg-Pro-Gln-Phe (chlorella-11) from *C. vulgaris* [87] and Leu-Asn-Gly-Asp-Val-Trp from *C. ellpsiodea* [88]. Chlorella-11 may be partially responsible for this anti-inflammatory activity as it has been found to reduce serum TNF-α levels and prostaglandin E2 (PGE2) production after lipopolysaccharide (LPS) activation in rats [89]. As IR is associated with low-grade inflammation, the anti-inflammatory activity of peptides and other components in *Chlorella* products may contribute to their antidiabetic effects. Meanwhile, the antihypertensive effects of peptidic fractions from *C. vulgaris* (200 mg/kg body weight) in spontaneously hypertensive rats has been attributed to oligopeptides such as the ACE inhibitors Ile-Val-Val-Glu (IC_50_ 315.3 µM), Ala-Phe-Leu (63.8 µM), Phe-Ala-Leu (26.3 µM), Ala-Glu-Leu (57.1 µM), and Val-Val-Pro-Pro-Ala (79.5 µM) [52].

### 4.4. Polyphenols from Chlorella

It has been suggested that phenolics in *Chlorella* preparations are partly responsible for their antidiabetic effects. It has been shown that *C. vulgaris* contains 5 mg rutin equivalents/g. The extracted phenolic fraction was shown to have α-amylase inhibitory activity (63.1% at 20 mg/L) [90]. A hydrophilic fraction obtained from fermented *C. vulgaris* biomass presented free radical scavenging activity (1,1-diphenyl-2-picrylhydrazyl free radical (DPPH) assay), antibacterial activity (against *Escherichia coli*, *Lactobacillus plantarum*, *Staphylococcus aureus*, and *Staphylococcus epidermidis*), and antifungal activity (against *Aspergillus niger*, *Candida albicans*, and *Saccharomyces cerevisiae*). The extract was shown to putatively contain polyphenols, as it was reactive in the Folin–Ciocalteu assay [91].

### 4.5. Vitamins and Minerals from Chlorella

*Chlorella* products contain vitamins B1, B2, B6, B12, niacin, folate (B9), biotin, pantothenic acid, C, D2, E, and K [16]. Of these, vitamins D2 and B12 are not found in plants.

*Chlorella* (*C. vulgaris*) contains high concentrations of folate (approximately 1.69–2.45 mg/100 g dry weight) [92]. Folate (vitamin B9) is a crucial nutrient involved in the synthesis of amino acids and nucleotides. There is some evidence that folate reduces the blood insulin levels in subjects with IR who are at risk of suffering from T2DM [93]. The capacity of folate to reduce the levels of T2DM-related homocysteine would explain this protection [94,95]. The main folate compounds in *Chlorella* products are 5-CHO-H_4_ folate (60–62%) and 5-CH_3_-H_4_ folate (24–26%), while the minor ones are 10-CHO-folate (5–7%), H_4_ folate (4%), and fully oxidized folate (3–6%) [92].

*C. stigmatophora* is particularly rich in the antioxidant vitamin E (669 mg/kg dry weight), whose antioxidant potential may contribute to protection against T2DM [96].

The high contents of minerals such as selenium in *Chlorella* in collaboration with its anti-inflammatory and antioxidative capacity may reduce fasting blood glucose and improve glycemia in view of the association between serum selenium and diabetes [97].

### 4.6. Carotenoids from Chlorella

*C. zofingiensis* is a source of astaxanthin, a xanthophyll carotenoid with health-promoting properties including, among others, the amelioration of chronic inflammatory diseases, MS, diabetes, diabetic nephropathy, and CVD [98,99]. In clinical studies, oral administration of astaxanthin (8 mg/day for 8 weeks) to patients with T2DM significantly reduced plasma glucose concentrations [100]. It is believed that the antidiabetic effect of astaxanthin is mainly mediated by its antioxidant activity, as reactive oxygen species (ROS) are key factors in the inducement of pancreatic β-cell damage by hyperglycemia [101]. The radical scavenging capacity of astaxanthin is common among unsaturated lipids and comes from its conjugated double bonds.

Compared with other carotenoids, astaxanthin is particularly well incorporated into the lipid bilayer of cellular membranes, where it prevents lipid oxidation without altering the lipid bilayer [102]. Astaxanthin has also been found to regulate intracellular oxidative stress by activating the MAPK, PI3K/Akt, and nuclear factor erythroid 2-related factor 2 (Nrf2)/antioxidant response element (ARE) signaling pathways in STZ-induced diabetic rats [103]. Astaxanthin also modulates the inflammatory response by inhibiting the release of pro-inflammatory cytokines such as interleukin-1B (IL-1B), interleukin-6 (IL-6), intercellular adhesion molecule-1 (ICAM-1), TNF-α, and monocyte chemoattractant protein-1 (MCP-1) [104]. Furthermore, astaxanthin appears to protect pancreatic β-cells from deterioration and death in both types of diabetes through a combination of interrelated antioxidant and anti-inflammatory effects [101]. The major source of astaxanthin is *Haematococcus pluvialis* (4–5% of cell dry weight) [105]. *C. zofingiensis* is an alternative source of carotenoids including astaxanthin (0.1–1% of cell dry weight) [99].

## 5. Other Emerging Microalgae

### 5.1. Tetraselmis

*Tetraselmis* contains flagellated species that form motile colonies found around the globe in both marine and water ecosystems [10]. An initial assessment report by *Agencia Española de Seguridad Alimentaria y Nutrición* (AESAN) for marketing dried *Tetraselmis chuii* was acknowledged by the EC in 2014 (https://ec.europa.eu/food/sites/food/files/safety/docs/novel-food_authorisation_2014_auth-letter_tetraselmis_chuii_en.pdf, accessed on 1 September 2020). This is an important step towards diversifying the list of approved microalgae for human nutrition. *Tetraselmis* species are sources of PUFAs, vitamin E, carotenoids, chlorophyll, tocopherols, and polyphenols [106] and can be used as probiotics in aquaculture [107]. *T. suecica* contains 74 g PUFAs, 70 g α-tocopherol, 3 g β-carotenoid, and 4 g polyphenols per kg biomass [106]. The possible application of *Tetraselmis* species in the prevention of diabetes has not been documented in the scientific literature; however, a patent application describes the activity of *T. suecica* in the prevention of obesity and diabetes via promoting the absorption of glucose into cells [108].

### 5.2. Isochrysis Galbana

*Isochrysis galbana* is a marine-based microalga pertaining to the family Isochrysidaceae within the phylum Haptophyta, which typically uses oil droplets for energy storage [10]. *Isochrysis galbana* is composed of 27% protein, 34% carbohydrates, and 11% fat [109]. It is a rich source of chlorophyll, carotenoids (fucoxanthin, β-carotene, diadinoxantin and diatoxantin), and chrysolaminarin (a polysaccharide made from glucose moieties linked by type β-1,3 and type β-1,6 glycosidic bonds) [109]. A total of 1.35% of the fat is alpha-linolenic acid and 37.1% is oleic acid (18:1 n-9), which is more than that present in soybean oil [109]. According to other reports, the oleic acid content is lower (15%) and the DHA content is 9–13% of the total fatty acid content [110,111]. Other studies have reported higher concentrations of EPA than DHA [11]. The proportions of EPA and DHA in *Isochrysis* preparations may vary considerably between species or strains, depending on the growing conditions [112]. The general consensus in the literature is that *Isochrysis* spp. and strains contain particularly large amounts of ω-3 PUFAs. One gram of *I. galbana* biomass may contain as much as 40 mg EPA + DHA [111].

*I. galbana*, as with *Tetraselmis suecica* and *C. stigmatophora*, is particularly rich in lipid-soluble (A and E) and B-group vitamins (including vitamins B1, B2 (riboflavin), B6 (pyridoxal), and B12) [96]. *I. galbana* also contains mono and digalactosyldiacylglycerols with highly unsaturated acyl chains that exhibit anti-inflammatory activity in vitro [113]. The differences in fatty acid composition between reports may be due to differences in the preparation of the *I. galbana* biomass analyzed, which is not always detailed in the literature. Apart from chrysolaminaran-like polymers [114], *I. galbana* contains other complex unspecified polysaccharides. The total phenolic content of *I. galbana* T-ISO was found to be 3 mg gallic acid equivalents per gram microalgae biomass [111].

A concentrated preparation of *I. galbana* biomass (50 mg/day) had an effect on alloxan monohydrate-treated male Sprague-Dawley rats. It decreased glucose, TAG, and TC levels and increased lactic acid bacteria (LAB) populations, resulting in minor signs of intestinal inflammation [115]. The effects on body mass were attributed to polysaccharides, which affect satiety and, consequently, energy intake and body composition. Cell wall polysaccharides are also believed to be involved in the blood glucose lowering effect. Again, the overall trend of lower TC and TAG and TC levels in the *I. galbana* diabetic group may be due to reductions in intestinal absorption caused by the polysaccharides acting as dietary fiber and lowering energy intake [116]. A histology exam showed that *I. galbana* had no negative effects on the gastrointestinal tract, in contrast to the alterations triggered by other microalgal species. The effects on microbiota were attributed to the constituent polysaccharides, as laminaran increased cecal weight, cecal SCFAs (acetic acid and propionic acid), and the ratio of bifidobacteria to total viable cells [117]. *I. galbana* has shown no adverse effects in rats [115,118].

### 5.3. Nannochloropsis

*Nanochloropsis* spp. are mostly marine microalgae pertaining to the phylum Ochrophyta. Like *Isochrysis*, they have the capacity to store energy in the form of lipid droplets [10]. *Nannochloropsis* spp are also rich sources of high-quality protein (30%), carbohydrates (10%), and fat (22%) [119]. The level of fat in *Nannochloropsis* biomass may be as high as 39% [57]. *Nannochloropsis* contains EPA as the major PUFA [120,121].

*N. gaditana* was shown to reduce blood insulin and HbA1c and attenuate oxidative stress and inflammation in STZ-induced diabetic male Wistar rats (10% in feed) [122]. In the same animal model of T1DM, *N. oculata* reduced blood glucose and concentrations of glucose, TC, TAG, and LDL cholesterol while increasing the concentrations of insulin and HDL cholesterol [123]. These results are in contrast with a previous study cited above that examined the effects of *I. galbana* and *N. oculata* on alloxan-induced diabetic male Sprague-Dawley rats [115]. In that study, *N. oculata* did not modify the altered concentrations of blood glucose, TC, or TAG [115]. The differences between *I. galbana* and *N. oculata* in that model may be related to differences in the polysaccharide content of the biomasses, as the effects of the former have been attributed to the prebiotic function of its cell wall components [115]. The hypoglycemic effect of *Nannochloropsis* in the STZ model might be related to its insulin increasing action. *N. oculata* was not found to have any adverse effects in rats [115]. A lipid extract from *Nannochloropsis* biomass was found to reduce plasma and liver cholesterol in rats fed a high-cholesterol diet [124].

Protein hydrolysates of *N. oculata* contain ACE inhibitory peptides (Leu-Glu-Gln and Gly-Met-Asn-Asn-Leu-Thr-Pro) with antihypertensive activity. The chemical composition of the crude hydrolysate is 31.0% protein, 1.3% lipids, 17.8% carbohydrates, and 4.4% fiber [125].

*N. gaditana* also contains a large amount of folate (2080 μg/100 g dry biomass), comparable to that of *Chlorella* and much higher than that of *Arthrospyra* [92]. As commented on before, folate offers protection against T2DM [93].

*Nannochloropsis* spp. contain phenolic acids (chlorogenic, caffeic, gallic, protocatechuic, hydroxybenzoic, syringic, vanillic, and ferulic acids) with free radical scavenging capacity, antifungal activity, and other in vitro activities [57,126]. These phenolics may contribute to the putative protective effects of the whole biomass against diabetes risk factors.

Polysaccharides of the -glucan type have been quantified (4.2% dry weight) in *N. salina* [80]. Thus, *Nannochloropsis* may be a good source of saccharides with prebiotic potential against IR.

Other emerging microalgae species may prove to be introduced as foodstuffs with functional properties. *Euglena gracilis* is also a rich source of protein, vitamins, lipids and paramylon, which is a β-1,3-glucan with immunostimulant activity that is only found in euglenoids [127]. A powdered preparation of *Euglena gracilis* was shown to reduce hyperglycemia and decrease food intake, body weight gain, and abdominal fat in Otsuka Long–Evans Tokushima fatty (OLETF) rats, which are another model of T2DM. Paramylon did not show any effect in this model [128].

## 6. Conclusions and Final Remarks

Microalgae are a low-fat, rich source of high-quality protein and bioactive functional components, such as polysaccharide fibers, polyphenols, carotenoids, phycobiliproteins, vitamins, sterols, and, particularly, PUFAs. *Spirulina* (*Arthrospira*), *Chlorella*, and *Tetraselmis* have been authorized for human consumption by the European Food Safety Authority (EFSA) and other regulatory agencies. Other microalgae, such as *Isochrysis galbana* and *Nannochloropsis* spp., have a long history of use in aquaculture and are candidates for use in human nutrition. Purified functional components such as -3 PUFAs (EPA/DHA) and astaxanthin are also used as food ingredients and supplements.

Supplementation of humans and other mammals with chlorella has been associated with antioxidant, antidiabetic, immunomodulatory, antihypertensive, and antihyperlipidemic effects, while supplementation of diabetic patients with spirulina has yielded contradictory results. The biomasses of *Chlorella*, *Arthrospyra*, *Tetraselmis*, *Isochrysis*, and *Nannochloropsis* contain components that may be effective against the different MS factors.

*Chlorella* spp. contain prebiotic polysaccharides such as β-glucans and arabinomannans (oligomers of arabinose and mannose) as well as other less known structures that exhibit hypolipidemic activity. Spirulina biomass has been shown to have some prebiotic effects (promotion of *Lactobacillus* and *Bifidobacterium*), but the putative active components have not been characterized as well as those in *Chlorella*. Chrysolaminarin is the major polysaccharide in *Isochrysis*.

Meanwhile, both *Chlorella* and *Arthrospyra* contain phenolic compounds with radical scavenging capacity and α-amylase/glycosidase inhibitory activity, two activities that may be behind the antioxidant, anti-inflammatory, and antihyperglycemic actions of the whole biomass. Little is known about the structure/activity relationship of microalgal polyphenols. The active components might be small phenolics rather than polymers. Peptides released by enzymatic hydrolysis of chlorella and spirulina biomass present hypoglycemic, anti-inflammatory, and antihypertensive activity. ACE inhibitors appear to be ubiquitous and effective at lowering diet-induced hypertension. Other components that may contribute antioxidant and anti-inflammatory activities to the protective effects against MS include vitamins (e.g., vitamin B9 or folate from *Chlorella*), carotenoids (e.g., astaxanthin from *Chlorella* and fucoxanthin from *Isochrysis*), and pigments/pigment proteins (e.g., phycobilins and phycocyanin from *Arthrospyra*). *Isochrysis* and *Nannochloropsis* are particularly rich in -3 PUFAs (EPA/DHA) with anti-inflammatory activity. The proportions of EPA and DHA may vary considerably among species or strains and depend on the growing conditions.

The mechanisms by which chlorella and spirulina might exert protective effects against diabetes and its risk factors in humans are largely unknown. Studies involving animal models have been limited to T1DM models such as alloxan- and STZ-induced T1DM in rodents. In such models, the microalgae are tested after the pancreatic function has already been severely affected. The results suggest that microalgae improve insulin sensitivity or protect β-cell function from oxidative stress and inflammatory damage. We have not found any mechanistic studies involving murine models of diet-induced MS or T2DM, that is to say, in a situation starting with IR which later progresses to pancreatic damage.

The composition of edible microalgae suggests they may be effective at two levels: in the early stages of IR development and in the later stages when pancreatic -cell function is already compromised. The early active components might be prebiotic polysaccharides, probably via the preservation of eubiosis and gut integrity. Then, antioxidant scavengers and anti-inflammatory lipid mediators such as carotenoids and -3 PUFAs (EPA/DHA) may counteract systemic inflammation and pancreatic damage. Phenolics may also act at different levels and stages—directly, by acting on gut microbiota, or indirectly, by stimulating the endogenous antioxidant defense system. ACE inhibitors and other peptides may contribute to antihypertensive and anti-inflammatory activity at the late stages of MS development.

These hypotheses need to be confirmed by new mechanistic studies in animal models. Microalgae are one of the foods of the future, as they are a source of high-quality protein and include an array of bioactive molecules with potential to act against the modern epidemics of obesity and diabetes while decisively contributing to a sustainable world through carbon dioxide fixation and the minimization of agricultural land use.

## Figures and Tables

**Table 1 nutrients-13-00563-t001:** Macronutrients and bioactive compounds present in edible microalgae (by wet weight).

Bioactive Compounds	*Arthrospira*	*Chlorella*
Proteins	60–70%	55–60%
Lipids	5% (ALA)	>10% (ω-3 PUFAs)
Dietary fiber	2%	>30%
Minerals	10% (P, Mg, K, Ca, Fe, Zn)	>10% (Fe, Ca, Mg, K, Zn)
Vitamins	E, B1, B2, B3, B9	C, E, K1, B12, B1, B2, B6
Pigments	Phycocyanine, carotenoids	1–4% chlorophylls, carotenoids

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
