# Peer review of "Edible Microalgae and Their Bioactive Compounds in the Prevention and Treatment of Metabolic Alterations"

_nutrients, 2021, doi:10.3390/nu13020563_

Round 1
Reviewer 1 Report
- Remove words like ‘probably’ Line 15, Line 250, Line 60 , ‘somehow’ line 64
- Line 16 – 20 Long sentence – reword
- Line 31 reconsider use of word ‘should’
- Phrasing Line 37 ‘incorporation in not to’
- Line 47 – resulting from ‘the’ total or partial lack...
- Line 93 - 96 Reword/expand upon as they are not subject to the Novel Food Regulation
- Line 122 – 127 long sentence, rephrase
- 127-130 – at what concentration? Was the extract characterised?
- Again 133-136 ‘administration of A. maxima’ – in what form? concentration? I would suggest providing this information when citing such studies; again similarly in line 140-142.
- Line 142-147 – first mention of changes in microbial populations and then maintenance of balanced gut flora is mentioned again/discussed in lines 191-194. I would introduce the maintenance of gut flora earlier in line 142-147, provides more understanding to the reader as to why a reduction in proteobacteria is a good finding or simply reword.. ‘reduced the population of fecal ‘pathogenic’ proteobacteria and explain the significance of being able to modulate the Firmicutesto Bacteroidetes ratio (F/B ratio)
- Lines 147-152 again what was the dosage or range of dosages in these clinical trials and lines 151-152 - expand upon what the authors are suggesting i.e...spirulina may be an adjuvant in treatments when taken as …a xx mg powdered daily supplement..
- Lines 147-152 Question – for the studies highlighted, who were the subjects i.e. healthy subjects or subjects with underlying health conditions, this is important when making conclusion statements regarding the benefits of incorporating algae in human diet..
- Lines 275-278 explain the significant of having potent α-glucosidase inhibitory activity – i.e. a management strategy of diabetes mellitus is to retard absorption of glucose by inhibition of carbohydrate hydrolyzing enzymes
- Lines 291 – what concentration of phycocyanin and how was it administered?
- Line 334 – typo ‘gens’
- Line 379 – I would expand line 379 i.e oral glucose tolerance test is the gold standard for diagnose of diabetes and possible expand upon your statement lines 383 to 385 , explain why this shift in microbes important? These microbes may not be immediately familiar to the reader
- Line 416 -417– I don’t like this sentence.
- Lines 489-490 how such C. zofingiensis compare to Haematococcus in terms of astaxanthin yield?
- Lines 494-497 – rephrase to clarify what you wish to say/convey e.g. in 2013 an initial assessment report by AESAN Scientific Committee concluded that the dried microalgae Tetraselmis chuii meets the criteria for acceptance as a novel food. This is great step towards utilising a greater diversity of algal species as a source of human food/nutrition.
Overall I found this to be a nicely prepared review and could benefit from a little extra detail here and there as outlined above.
Author Response
Thank you for all the suggestions. Following the reviewer indications, all proposed changes has been modified, corrected or rephrased accordingly.
127-130 – at what concentration? Was the extract characterised?
This information is now included in the paper (lines 110-112).
Again 133-136 ‘administration of A. maxima’ – in what form? concentration? I would suggest providing this information when citing such studies; again similarly in line 140-142.
This information is now included in the manuscript (lines 115-118)
Line 142-147 – first mention of changes in microbial populations and then maintenance of balanced gut flora is mentioned again/discussed in lines 191-194. I would introduce the maintenance of gut flora earlier in line 142-147, provides more understanding to the reader as to why a reduction in proteobacteria is a good finding or simply reword.. ‘reduced the population of fecal ‘pathogenic’ proteobacteria and explain the significance of being able to modulate the Firmicutesto Bacteroidetes ratio (F/B ratio)
Now this is clarified in lines 122-126.
Lines 147-152 again what was the dosage or range of dosages in these clinical trials and lines 151-152 - expand upon what the authors are suggesting i.e...spirulina may be an adjuvant in treatments when taken as …a xx mg powdered daily supplement..
The kind of spirulina preparation is not specified in the meta-analysis, nor is the dosage.
Lines 147-152 Question – for the studies highlighted, who were the subjects i.e. healthy subjects or subjects with underlying health conditions, this is important when making conclusion statements regarding the benefits of incorporating algae in human diet..
Certainly, this detailed information is of interest. The reader may consult the meta-analysis cited here. The idea of these sentence is to acknowledge the increasing consensus that Spirulina may reduce weight gain.
Lines 275-278 explain the significant of having potent α-glucosidase inhibitory activity – i.e. a management strategy of diabetes mellitus is to retard absorption of glucose by inhibition of carbohydrate hydrolyzing enzymes
This information is now explained in lines 229-231.
Lines 291 – what concentration of phycocyanin and how was it administered?
This information is included in lines 240-242.
Line 334 – typo ‘gens’
This is already corrected.
Line 379 – I would expand line 379 i.e oral glucose tolerance test is the gold standard for diagnose of diabetes and possible expand upon your statement lines 383 to 385 , explain why this shift in microbes important? These microbes may not be immediately familiar to the reader
The significance in the shifts of most of these microbes is not currently known, so we have just reported the results.
Line 416 -417– I don’t like this sentence.
We have modified the sentence because the peptides are mostly present as part of larger proteins and are released upon hydrolysis (lines 335-337).
Lines 489-490 how such C. zofingiensis compare to Haematococcus in terms of astaxanthin yield?
We have included the information requested. Astaxanthin content is one-foll lower in C. zofingiensis than in Haematococcus pluvialis (lines 392-394).
Lines 494-497 – rephrase to clarify what you wish to say/convey e.g. in 2013 an initial assessment report by AESAN Scientific Committee concluded that the dried microalgae Tetraselmis chuii meets the criteria for acceptance as a novel food. This is great step towards utilising a greater diversity of algal species as a source of human food/nutrition.
We have modified the paragraph to make it more clear and to underline the importance of the introduction of Tetraselmis for human consumption (lines 402-408).

Reviewer 2 Report
The manuscript by Ramos-Romero et al. is a comprehensive review article on the utilization of microalgae focused on prevention and treatment of human metabolic disorder. The article is well written and the review of the current literature on microalgal utilization is exhaustive.
I have several comments and questions below, which I hope will be helpful to the authors.
- For the sake of readers, it would be nice to have an explanation of each microalgae discussed in this paper. What phylum they belong to, where their main habitat is, etc.
- In the experiments on chlorella and spirulina, I think it would be better to clearly state in what state the chlorella and spirulina are used in the experiments. For example, are they dried cells, crushed cells, or extracts?  Which species of chlorella or spirulina did you use?
- In the section on unsaturated fatty acids, I think it would be more helpful to show the composition of the fatty acids or the names of the main fatty acids, rather than just a mixture of fatty acids.
- What are species of microalgae authorized for human consumption by the European Food Safety Authority (EFSA) other than Spirulina (Arthrospira), Chlorella (Chlorella spp.) and Tetraselmis spp.? I think it would be better to explain a little more why you have included Isochrysis galbana and Nannocloropsis spp. in this paper.
- For use as food, mass cultivation is necessary, but has mass cultivation technology been established for Tetraselmis, Isochrysis galbana and Nannocloropsis discussed in this paper?
- Are Tetraselmis, Isochrysis galbana and Nannocloropsis microalgae that we have some experience eating?
- I think Euglena gracilis is also a promising microalga, but why is it not discussed in this paper? Does it have little effect on human metabolic disorder?
- Lines 22-23; “Spirulina (Arthrospira spp.) and Tetraselmis spp., as well as Isochrysis spp. and Nannochloropsis spp.” should be “Spirulina (Arthrospira) and Tetraselmis spp., as well as Isochrysis spp. and Nannochloropsis spp.”
- Were the data in Table 1 measured in your laboratory or did you cite the data?
- Line 58; What does CDV stand for?
- Lines 74, 553, 556, 584, 590, 599; “Nannochloropsis spp.” should be “Nannochloropsis”
- Lines 88-89; “polyunsaturated fatty acids PUFAs” should be “polyunsaturated fatty acids (PUFAs)”.
- Line 101; “microelements” should be “minerals”.
- Lines 102-103; “polyunsaturated fatty acids (especially γ-linolenic acid, ALA)” should be “PUFAs (especially γ-linolenic acid (ALA, 18:3 n-6))”.
- Line 108; “non-alcoholic fatty liver disease” should be “non-alcoholic fatty liver disease (NAFLD)”.
- Line 122; “In vivo studies indicate that Arthospira maxima and A. platensis” should be “In vivo studies indicate that Arthospira maxima and platensis”.
- Line 128; “A. maxima” should be italic.
- Line 132; “AMPKα-induced” should be “adenosine 5’-monophosphate-activated protein kinase-α (AMPK-α)-induced”.
- Line 133; A. maxima” should be italic.
- Line 136; “TNF-α” should be “tumor necrosis factor alpha (TNF-α)”.
- Line 138; “AMP-activated protein kinase (AMPK)” should be “AMPK”.
- Lines 140 and 143; “A. platensis” should be italic.
- Lines 172-173; “tumor necrosis factor alpha (TNF-α)” should be “TNF-α”.
- Lines 251-252; “adenosine 5’-monophosphate-activated protein kinase-α (AMPK-α)” should be “AMPK-α”.
- Line 259; “serum triglyceride, total cholesterol” should be “serum TAG, total cholesterol (TC)”.
- Lines 260, 261, 274, 301, 302, 314, 317, 330, 332, 536, 542, 561, 562, 566, 573 and 574; “cholesterol” should be “TC”.
- Line 293; “total cholesterol (TC) levels and triglycerides (TG)” should be “total TC levels and TAG”.
- Lines 309, 390 and 399, ; “Chlorella spp.” should be “Chlorella ”.
- Line 315; “[62 ]” should be “[62]”.
- Lines 318, 561, and 566; “triglycerides” should be “TAG”.
- Line 323-324; “non-alcoholic fatty liver disease (NAFLD)” should be “NAFLD”.
- Line332; “total cholesterol and triglycerides” should be “total TC and TAG”.
- Line 380: About “the extract from platensis (SP55)”, I am wondering if SP55 means the ethanolic (55% ethanol) fraction extracted from S. platensis.
- Line 383; “The CP55 fraction” should be “CP55”.
- Lines 536 and 542; “triacylglycerol” should be “TAG”.
- Line 596; “Spirulina (Arthrospira spp.)” should be “Spirulina (Arthrospira)”.
- Line 493, 497, 502 and 597; “Tetraselmis spp.” Should be “Tetraselmis”.
- Lines 614-615; “Lactobacillus spp. and Bifidobacterium spp.” should be “Lactobacillus and Bifidobacterium spp.”.
- Lines 636-664; This paragraph may mislead the reader into thinking that Chlorella and Spirulina contain w-3 PUFAs (EPA/DHA). I don't think EPA/DHA is present in green algae or cyanobacteria.
- Lines 665 and 666; “S.R.R.” might be “S.R.-R.”.
- Line 666; “J.L:T” should be “J.L.T.”.
- References; Please align the style of References correctly. For example, in the title of the paper, only the word at the beginning of the title is capitalized in some cases and not in others. The names of the papers are not aligned. Some of them are abbreviated, some are not.
Author Response
Thank you for your revisions and comments.
For the sake of readers, it would be nice to have an explanation of each microalgae discussed in this paper. What phylum they belong to, where their main habitat is, etc.
We have supplemented the text with some additional information about the different microalgae reviewed (lines 89-90, 98-99, 399-400, 412-413, 447-448).
In the experiments on chlorella and spirulina, I think it would be better to clearly state in what state the chlorella and spirulina are used in the experiments. For example, are they dried cells, crushed cells, or extracts?  Which species of chlorella or spirulina did you use?
We have specified the particular species and the kind of preparation whenever the source provided the information. In most instances the preparation is dried crushed biomass after non-specified enzymatic treatment.
In the section on unsaturated fatty acids, I think it would be more helpful to show the composition of the fatty acids or the names of the main fatty acids, rather than just a mixture of fatty acids.
We have specified the names of the particular fatty acids only in the cases where the information was provided by the authors of the corresponding studies. In many instances the particular structures are not specified.
What are species of microalgae authorized for human consumption by the European Food Safety Authority (EFSA) other than Spirulina (Arthrospira), Chlorella (Chlorella spp.) and Tetraselmis spp.? I think it would be better to explain a little more why you have included Isochrysis galbana and Nannocloropsis spp. in this paper.
These three are the only microalgae authorised for human consumption. We have added a short extension of the arguments for including Isochrysis galbana and Nannochloropsis in the review (Lines 63-66).
For use as food, mass cultivation is necessary, but has mass cultivation technology been established for Tetraselmis, Isochrysis galbana and Nannocloropsis discussed in this paper?
Yes, particularly Isochrysis galbana and Nannocloropsis spp. for their use in aquaculture. As the goal of the review is the biological effects we have not concentrated in the production of the corresponding biomasses.
Are Tetraselmis, Isochrysis galbana and Nannocloropsis microalgae that we have some experience eating?
There is no information available on this, as far as we know.
I think Euglena gracilis is also a promising microalga, but why is it not discussed in this paper? Does it have little effect on human metabolic disorder?
This is an interesting suggestion for improvement of the review. We have introduced a comment on this microalga at the end of the main body and before the conclusions (lines 479-484).
Lines 22-23; “Spirulina (Arthrospira spp.) and Tetraselmis spp., as well as Isochrysis spp. and Nannochloropsis spp.” should be “Spirulina (Arthrospira) and Tetraselmis spp., as well as Isochrysis spp. and Nannochloropsis spp.”
Now this is corrected as suggested.
Were the data in Table 1 measured in your laboratory or did you cite the data?
This is now punctualized in the text.
Line 58; What does CDV stand for?
This was a mistake; it is already corrected.
Lines 74, 553, 556, 584, 590, 599; “Nannochloropsis spp.” should be “Nannochloropsis”
Lines 88-89; “polyunsaturated fatty acids PUFAs” should be “polyunsaturated fatty acids (PUFAs)”.
Line 101; “microelements” should be “minerals”.
Lines 102-103; “polyunsaturated fatty acids (especially γ-linolenic acid, ALA)” should be “PUFAs (especially γ-linolenic acid (ALA, 18:3 n-6))”.
Line 108; “non-alcoholic fatty liver disease” should be “non-alcoholic fatty liver disease (NAFLD)”.
Line 122; “In vivo studies indicate that Arthospira maxima and A. platensis” should be “In vivo studies indicate that Arthospira maxima and platensis”.
Line 128; “A. maxima” should be italic.
Line 132; “AMPKα-induced” should be “adenosine 5’-monophosphate-activated protein kinase-α (AMPK-α)-induced”.
Line 133; A. maxima” should be italic.
Line 136; “TNF-α” should be “tumor necrosis factor alpha (TNF-α)”.
Line 138; “AMP-activated protein kinase (AMPK)” should be “AMPK”.
Lines 140 and 143; “A. platensis” should be italic.
Lines 172-173; “tumor necrosis factor alpha (TNF-α)” should be “TNF-α”.
Lines 251-252; “adenosine 5’-monophosphate-activated protein kinase-α (AMPK-α)” should be “AMPK-α”.
Line 259; “serum triglyceride, total cholesterol” should be “serum TAG, total cholesterol (TC)”.
Lines 260, 261, 274, 301, 302, 314, 317, 330, 332, 536, 542, 561, 562, 566, 573 and 574; “cholesterol” should be “TC”.
Line 293; “total cholesterol (TC) levels and triglycerides (TG)” should be “total TC levels and TAG”.
Lines 309, 390 and 399, ; “Chlorella spp.” should be “Chlorella ”.
Line 315; “[62 ]” should be “[62]”.
Lines 318, 561, and 566; “triglycerides” should be “TAG”.
Line 323-324; “non-alcoholic fatty liver disease (NAFLD)” should be “NAFLD”.
Line332; “total cholesterol and triglycerides” should be “total TC and TAG”.
All the previous recommendations have been now included in the text.
Line 380: About “the extract from platensis (SP55)”, I am wondering if SP55 means the ethanolic (55% ethanol) fraction extracted from S. platensis.
This is now specified as reviewer has recommended.
Line 383; “The CP55 fraction” should be “CP55”.
Lines 536 and 542; “triacylglycerol” should be “TAG”.
Line 596; “Spirulina (Arthrospira spp.)” should be “Spirulina (Arthrospira)”.
Line 493, 497, 502 and 597; “Tetraselmis spp.” Should be “Tetraselmis”.
Lines 614-615; “Lactobacillus spp. and Bifidobacterium spp.” should be “Lactobacillus and Bifidobacterium spp.”.
All the previous recommendations have been now included in the text.
Lines 636-664; This paragraph may mislead the reader into thinking that Chlorella and Spirulina contain w-3 PUFAs (EPA/DHA). I don't think EPA/DHA is present in green algae or cyanobacteria.
Both Isochrysis and Nannochloropsis contain EPA/DHA. We do not think the reader will infer than Spirulina and Chlorella also do.
Lines 665 and 666; “S.R.R.” might be “S.R.-R.”.
Line 666; “J.L:T” should be “J.L.T.”.
These changes are already included.
References; Please align the style of References correctly. For example, in the title of the paper, only the word at the beginning of the title is capitalized in some cases and not in others. The names of the papers are not aligned. Some of them are abbreviated, some are not.
All the references have been revised and corrected.
